# Predicting stock returns using machine learning combined with data envelopment analysis and automatic feature engineering: A case study on the Vietnamese stock market

**Hoang Thanh Nhon[1], Nga Do-Thi[2], Thao Nguyen-Trang** [3,4]*

1 Faculty of Commerce, Van Lang University, Ho Chi Minh City, Vietnam, **2** Account Manager, VNPT Bussiness Center, Ho Chi Minh City, Vietnam, **3** Laboratory for Artificial Intelligence, Institute for Computational Science and Artificial Intelligence, Van Lang University, Ho Chi Minh City, Vietnam, **4** Faculty of Information Technology, Van Lang School of Technology, Van Lang University, Ho Chi Minh City, Vietnam

* thao.nguyentrang@vlu.edu.vn

**Data availability statement:** All data files are available from the link https://www.kaggle.com/datasets/thaonguyentrang/vietnam-stock-returns-with-efficiency-scores-dea.

## Abstract

In financial markets, predicting stock returns is an essential task for investors. This paper is one of the first studies using business efficiency scores calculated from data envelopment analysis to predict stock returns. In the meantime, this is also one of the first studies to comprehensively investigate the performance of machine learning models and automatic feature engineering techniques in the context of predicting returns in the Vietnamese stock market. Specifically, the data from 2019 to 2024 of 26 real-estate enterprises on Ho Chi Minh Stock Exchange are collected. Based on relevant technical indicators, fundamental indicators, and business efficiency scores, a comparison of various machine learning models' performance is provided. The results indicate that incorporating business efficiency scores significantly enhances the models' accuracy. For example, the deep neural network model shows a decrease in RMSE from 0.926 to 0.375, MAE from 0.337 to 0.196, and MAPE from 134.63 to 114.71. Furthermore, the gradient boosted tree model, when combined with business efficiency scores and automatic feature engineering, achieves the best results, yielding an MAE of 0.122 and an MAPE of 103.19. The obtained results reveal a significant improvement in terms of accuracy when using the business efficiency score with the automated feature engineering technique.

## 1 Introduction

In the financial domain, forecasting quarterly stock returns is of great importance. Investors and analysts constantly seek signals that can accurately reflect price trends to optimize investment decisions and minimize risks. The rapid advancement of technology has driven

**Funding:** The author(s) received no specific funding for this work.

**Competing interests:** The authors have declared that no competing interests exist.

the transition from traditional methods to machine learning models, enhancing the effectiveness of analysis and prediction.

The Vietnamese stock market is a young and developing market. This market has low liquidity and is strongly influenced by political factors. These factors can lead to market specificity, making models that are good for other markets may not be good when applied to the Vietnamese market. Previous studies have mainly focused on predicting market indicators, such as VNIndex and VN30, and often use univariate time series analysis techniques [1–3]. However, the application of machine learning methods to forecast Returns based on technical, fundamental factors, and especially efficiency scores from data envelopment analysis (DEA) has not been done yet. Therefore, this study will compare machine learning methods with a variety of independent variables to select the most suitable machine learning model to forecast the rate of return for the Vietnamese market. *The selection of a suitable model for local characteristics is the third contribution of this paper.* The paper's contributions are summarized as follows.

- This paper is among the first studies to use the efficiency score calculated by DEA as a predictor for stock return prediction.
- This paper pioneers the application of automatic feature engineering to explore latent and essential predictors related to the efficiency score and other variables, thereby enhancing the model's performance.
- This paper serves as a valuable reference for selecting an appropriate machine learning model to predict stock returns in the Vietnamese market based on the efficiency score.

The following sections of this paper are organized as follows. First, in Sect 2, we review the literature and background theories related to the research topic. Then, in Sect 3, we present the research methods. In Sect 4, we report the experimental results. In Sect 5, we draw conclusions.

## 2 Literature review

### 2.1 Application of machine learning in return prediction

Machine learning models typically utilize two groups of independent variables: technical and fundamental. Technical variables, such as RSI, MACD, or support and resistance levels, rely heavily on historical stock price data to identify market trends and dynamics. Their advantage lies in promptly capturing price movements; however, they often overlook fundamental factors that affect a stock's intrinsic value. In contrast, fundamental analysis focuses on the financial health of firms, making it more suitable for medium- to long-term forecasting, spanning several quarters to a few years.

In the early stages of research on the application of machine learning to stock selection, the work of Graham [4] had the most profound influence on subsequent studies. In 1999, Quah et al. developed a model utilizing a Feedforward Neural Network (FNN) to anticipate stock prices based on quarterly fundamental financial indicators [5]. In 2008, Quah [6] applied a neural network model with fundamental financial indicators as inputs and the Dow Jones Industrial Average (DJIA) as the output. This research aimed to enhance decision-making processes, providing valuable insights for portfolio management.

Numerous scholars have applied machine learning models to predict stock prices. Yu et al. [7] introduced a hybrid model between Support Vector Machine (SVM) and Principal Component Analysis (PCA), while Zhang et al. [8] experimented with AdaBoost. Additionally,

Sabbar and El Kharrim [9] and Olorunnimbe and Viktor [10] have investigated the prediction of stock returns using deep learning models. Namdari and Li [11] employed an FNN model to predict stock price trends using data from 578 companies listed on Nasdaq from the second quarter of 2012 to the second quarter of 2017. The FNN model demonstrated superior performance compared to other models in predicting stock prices. Huang et al. [12] applied the Random Forest (RF) model to predict stock returns by analyzing 19 financial indicators. The selected stocks demonstrated superior performance, yielding significantly higher returns. Chen et al. [13] utilized a Graph Convolutional Neural Network (GC-CNN) to forecast stock prices. This model exhibited exceptional performance in predicting long-term stock price movements, showcasing its effectiveness in capturing complex patterns in financial data. Hong et al. [14] introduced a strategic neural network model focused on analyzing stock price volatility. This approach effectively reduced errors in stock market analysis, providing valuable support to investors in making informed investment decisions. Huang et al. [12] applied Random Forest (RF) model to predict stock returns by analyzing 19 financial indicators derived from fundamental analysis. Tsai et al. [15] focused on using machine learning techniques to predict stock returns in the Taiwan (TW) stock market, emphasizing the importance of combining both technical and fundamental variables. Olorunnimbe and Viktor [10] provided a comprehensive review of data types used in machine learning return forecasting, thereby confirming the role of both analytical methods in optimizing predictions. Some other studies can be listed as [16–19].

## 2.2 Research of DEA

Data Envelopment Analysis (DEA) is a method of measuring the performance of different production units or organizations [20]. DEA uses a mathematical model to compare the efficiency of these units based on their inputs and outputs. This method allows us to determine the level of efficiency compared to other units in the same field, thereby indicating the best-performing units and those that need improvement. DEA is widely applied in many fields, including education, health care, and manufacturing [21–23].

The application of DEA in the stock market has demonstrated its suitability as a method for analyzing efficiency and supporting decision-making [24–26]. DEA is widely used to assess the relative efficiency of investment entities such as mutual funds, investment portfolios, and listed companies by comparing multiple input factors (e.g., capital, risk level, costs) with output factors (e.g., returns, profitability, shareholder value).

For instance, Zohdi et al. [27] applied DEA to evaluate the performance of investment companies listed on the Tehran Stock Exchange, using financial ratios as inputs and outputs to measure their relative efficiency, thereby providing valuable insights for investors. Similarly, Roslah Arsad et al. [28] integrated DEA with DuPont analysis to assess the operational efficiency of companies and support stock selection, offering a more comprehensive perspective on financial performance.

In portfolio management, Hosseinzadeh et al. [29] demonstrated the utility of DEA in pre-selecting efficient assets, helping to optimize investment portfolios by narrowing down a large set of investment options to those with the best risk-return ratios. This method is particularly valuable for managing large-scale portfolios and reducing computational complexity. The integration of DEA with advanced analytical tools highlights its adaptability to modern financial challenges, such as processing massive financial datasets and incorporating multidimensional performance metrics.

Although the efficiency score is a valuable indicator, to date, no research has applied it as an independent variable in return forecasting. Therefore, *the first important contribution of*

*this paper is the use of the efficiency score as a predictor in a machine learning model.* By combining the efficiency score with technical and fundamental analysis variables, we hope to provide a new approach that provides deeper insights into the predictability of future stock returns.

## 2.3 Auto-feature engineering

Auto-Feature Engineering is a process of extracting features from raw data to improve the performance of predictive models. The advantage of auto-feature engineering is the ability to detect and optimize latent variables that humans may ignore, thereby improving the accuracy of the model.

In the field of stock prediction, a number of researchers have applied this technique to enhance predictor quality. Long et al. [30] applied deep learning as a feature engineering technique to financial time series for predicting stock price moments. Dami and Esterabi [31] applied Long Short-Term Memory (LSTM) deep neural networks to predict returns on the Tehran Stock Exchange, using AutoEncoder-based feature selection to reduce the number of input features. Yun et al. [32] proposed a hybrid GA-XGBoost prediction system with an advanced feature engineering process, including feature set expansion, data preparation, and optimal feature set selection. The experimental results show that this procedure significantly improves the accuracy of stock price direction prediction. Other related studies can be referred to in [33–37].

It can be seen that feature engineering has been widely applied in the field of financial market prediction. However, most studies focus on time series transformations through Wavelet, Fourier, or deep learning-based transformations. Automatic feature engineering for panel data is still relatively limited. In addition, studies focus on stock moment prediction, and no study has applied automatic feature engineering for return prediction, especially with the original variable set containing efficiency scores calculated from DEA. Therefore, this paper applied an auto feature engineering technique to the input set, including technical index, fundamental index, and efficiency score, to discover the important latent variables and select the most important variables in predicting return.

## 3 Data and methodology

### 3.1 Data

The financial report data for all real estate companies examined in this study were collected from the Ho Chi Minh City Stock Exchange (HOSE). To date, there are approximately 26 real estate companies listed on the Ho Chi Minh City Stock Exchange. These stocks were selected based on their market capitalization rankings on the exchange. We gathered financial reports for all listed companies from the third quarter of 2019 to the second quarter of 2024. From these financial reports, we calculated 12 financial indicators, which were used as input data for the machine-learning models ($X_1$ to $X_{12}$ in Table 1). Additionally, we applied Data Envelopment Analysis (DEA) to compute the efficiency score ($H$ in Table 1), which also served as an input variable for the machine learning models. Finally, we calculated the rate of return ($Y$), which was used as the output variable of the machine learning models. For reference, we have published the detailed dataset at the following link: "https://www.kaggle.com/datasets/thaonguyentrang/vietnam-stock-returns-with-efficiency-scores-dea".

**Table 1.** The variables used and equations.

| Variables | Description | Equation |
|---|---|---|
| $X_1$ | Current Ratio | Current assets/Current liabilities |
| $X_2$ | Debt to Equity | Long Term Debt/Shareholders' Equity |
| $X_3$ | Long-term Debt to Total Capital | Long Term Debt/(Shareholders' Equity + Long Term Debt) |
| $X_4$ | Account Receivable Turnover | Credit Sales/Account Receivables |
| $X_5$ | Inventory Turnover | Total Sales/Inventories |
| $X_6$ | Asset Turnover | Total Sales/Total Assets |
| $X_7$ | Return on Equity | Net Income/Shareholders' Equity |
| $X_8$ | Return on Asset | Net Income/Total Assets |
| $X_9$ | Gross Margin | (Total Sales - Cost of Goods Sold)/Total Sales |
| $X_{10}$ | Operating Margin | Operating Income Before Interest and Taxes/Total Sales |
| $X_{11}$ | Pretax Margin | Operating Income Before Taxes/Total Sales |
| $X_{12}$ | Net Margin | Net Income/Total Sales |
| $H$ | Business efficiency | Calculated from DEA model |
| $Y$ | Return | (Stock price at $t$ - Stock price at $t-1$)/Stock price at $t-1$ |

## 3.2 Data envelopment analysis

DEA is one of the typical techniques to evaluate the effectiveness of a set of objects with homogeneous characteristics (Decision-Making Units - DMUs). This technique examines the relationship between input and output variables to determine efficiency scores so that an object or a DMU can obtain a higher efficiency score when it achieves higher outputs and consumes similar inputs, or when it achieves similar outputs and consumes lower inputs, compared to the other DMUs.

In order to properly apply DEA in various areas, a number of variant models have been proposed, such as CCR (Charnes, Cooper, Rhodes), BCC (Banker, Charnes, Cooper), SBM (slacks-based measure), and so forth [20,38,39]. Among the mentioned models, CCR is the most popular one. According to this model, the following optimization problem can be used to determine the efficiency score $H_i$ of a decision-making unit $DMU_i$.

$$\text{maximize } H_i = \frac{\sum_{j=1}^{n} w_{ij} s_{ij}}{\sum_{l=1}^{m} u_{il} r_{il}},$$

$$\text{subject to } \frac{\sum_{j=1}^{n} w_{ij} s_{kj}}{\sum_{l=1}^{m} u_{il} r_{kl}} \leq 1, \forall k = 1, 2, \dots N,$$

$$\tag{1}$$

where $m$ represents the number of inputs, $n$ represents the number of outputs, $r_{il}$ represents the input $l$ of $DMU_i$, $s_{ij}$ represents the output $j$ of $DMU_i$, and $N$ represents the total number of DMUs. The weight vectors that must be identified for the $i$-th optimization problem are $\mathbf{u}_i = (u_{i1}, \dots, u_{im})$ and $\mathbf{w}_i = (w_{i1}, \dots, w_{in})$.

In this study, each enterprise is considered a DMU. Each DMU is evaluated through multiple input variables and multiple output variables simultaneously. The input variables include average equity and total assets, whereas the output variables include net income, operating profit, and revenues. The business efficiency score H determined by Eq (1) is then further considered a predictor in the machine learning model.

## 3.3 Machine learning models

**3.3.1 Generalized linear model.** The Generalized Linear Model (GLM) is an extension of the conventional linear regression model. The GLM is able to model different types of data, such as binary, count, and continuous data, through different hyperparameters related to error distributions and link functions. This allows the GLM to solve problems with various data distribution. The general equation of the GLM is presented as follows.

$$g(E(Y)) = \beta \mathbf{X}, \tag{2}$$

where $g$ is the link function, $E(Y)$ is the expectation of the response variable, $\mathbf{X}$ is the matrix of predictors, and $\beta$ is the vectors of regression coefficients.

**3.3.2 Neural networks and deep neural networks.** The Neural Networks (NNs) model is inspired by the reasoning activity of the human brain. An NNs model is composed of components: nodes (neurons), layers, weights, and activation functions. The layers of a network usually include an input layer, a hidden layer, and an output layer. We often distinguish between an NNs with one hidden layer (Shallow NNs or NNs ), and an NNs with several hidden layers (Deep NNs or DNN). An illustration for calculating a prediction value $\hat{y}$ based on an NNs model consisting of 1 hidden layer with $n$ nodes, $d$ predictors $x_i$ is depicted in Eq (3).

$$\hat{y} = f^{(1)} \left( \sum_{j=1}^{n} \alpha_j^{(1)} f_j^{(0)} \left( \sum_{i=1}^{d} \alpha_{ij}^{(0)} x_i + \beta_j^{(0)} \right) + \beta^{(1)} \right), \tag{3}$$

where $\alpha_{ij}^{(0)}$ and $\beta_j^{(0)}$ are the weights that combine and forward information from the input layer to the hidden layer; $\alpha_j^{(1)}$ and $\beta^{(1)}$ are the weights that combine and forward information from the hidden layer to the output layer; $f_j^{(0)}$ and $f^{(1)}$ are the activation functions. In Eq (3), the weights are optimized by gradient-based optimization during backpropagation.

**3.3.3 Decision tree.** Regression Tree is a popular and flexible machine learning method applied to classification and regression [40–42]. The objective of employing a tree algorithm is to predict the value of a target variable by acquiring basic decision rules from data features. This model employs a tree-like structure in decision-making, beginning with a root node and dividing the tree into branches and leaves. The Decision Tree algorithm is summarized by the subsequent procedures.

1. Place the best variable in the data set with the highest predictive ability at the root node.
2. Split the data set into subsets based on the root node. The subsets should contain data with the same value condition for the root node and receive the same predicted value.
3. Continue building the tree by repeating steps 1 and 2 until it satisfies the condition of the minimum number of leaves or the maximum depth of the tree.

In the steps above, the conditions used to split the tree into subsets play an important role. Here, we split the tree based on the criterion of minimum Sum of Squared Errors (SSE).

$$SSE = \Sigma_{i \in S_1} (y_i - \bar{y}_1)^2 + \Sigma_{j \in S_2} (y_j - \bar{y}_2)^2,$$

where $y_i$ is the actual value of observation $i$ in subset $S_1$, $y_j$ is the actual value of observation $j$ in subset $S_2$, and $\bar{y}_1$ and $\bar{y}_2$ are the average outputs in the subsets $S_1$ and $S_2$.

**3.3.4 Random forest.** The Random Forest (RF) generates a number of regression trees trained based on different subsets of the original data. Specifically, the subsets are drawn

through a random sampling process with replacement, and then an aggregation rule is utilized to make a final prediction. The use of multiple viewpoints of individual trees to conclude is a critical point to make the RF have a high level of generality, low bias, and low variance. The RF is summarized in the following steps. An illustration of how RF works is depicted in Fig 1.

1. Create $n$ samples based on $N$ observations in the training set, using the random sampling with replacement method.
2. Randomly select $k$ variables based on $p$ variables in the training set ($k < n$).
3. Establish a regression tree based on the data generated from Step 1 and Step 2.
4. Repeat the three steps above $T$ times to establish $T$ regression trees.
5. Aggregate the results by averaging the prediction of $T$ trees.

Gradient Boosted Trees (GBT) is an ensemble learning approach where a regression tree is built and gradually updated to boost the model performance [43]. Specifically, the GBT starts with building a simple tree based on the training data. Then, a new regression tree is built to predict the difference between the actual value and the predicted value of the previous tree. This process is repeated many times to reduce the prediction error. GBT typically uses gradient descent to optimize the loss function during training. By combining multiple weak regression trees into a single strong model, GBT is able to provide a higher performance compared to others. An illustration of how GBT works is depicted in Fig 2.

### 3.4 Automatic feature engineering

In order to generate and select the most relevant variables, automatic feature engineering is utilized. This technique can automatically generate new variables based on existing variables and then select variables into the model based on a multi-objective optimization algorithm [44,45]. Some of the commonly used functions in automatic feature engineering include addition, subtraction, multiplication, division, inverse, radical, exponent, logarithm, absolute value, sign, min, max, and combinations of the above functions. Further details of the used functions can be found in Table 2.

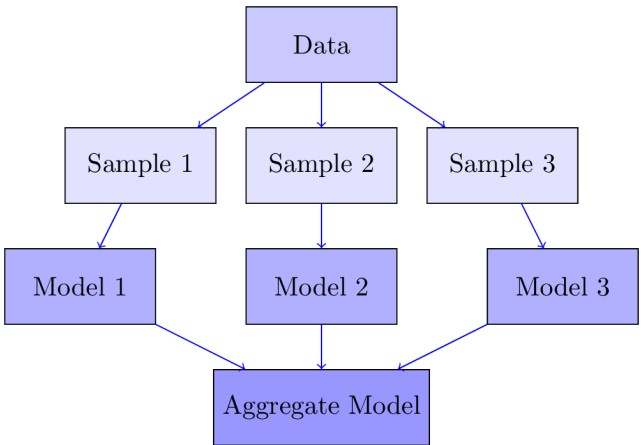

**Fig 1. An illustration of how RF works.**

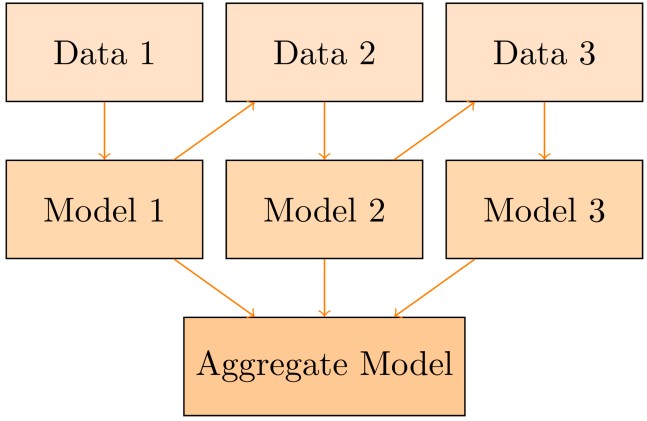

**Fig 2. An illustration of how GBT works.**

**Table 2. Details of the used functions in automatic feature engineering.**

| Origin | Generation | Description |
|---|---|---|
| $x$ | $\log(x), \sqrt{x}, \mathrm{sgn}(x), x^a, 1/x, |x|$, etc. | Apply unary functions to a single input to capture non-linearity or scale normalization |
| $x,y$ | $x+y, x-y, xy, x/y, \min(x,y), \max(x,y)$, etc. | Binary combinations of features using basic arithmetic or extremum operators |
| $x,y,z$, etc. | $f_1(f_2(x)), \log(x/y), \log(x)(y+\sqrt{z})$, etc. | Combinations of multiple functions and variables to model complex, hierarchical, or interaction effects |

Based on the generated variables, we can not only enhance the model accuracy but also detect possible nonlinear relationships and interactions between variables. To optimize variables, the research focuses on model complexity and performance. Complexity is calculated based on the number of variables; each variable contributes one to the overall complexity. Additional functions, such as $\log(x)$, are counted as one in the complexity measure. Meanwhile, model performance is measured using MSE. Based on the Pareto frontier, we can identify variables that optimally align with the objectives when constructing the model. This method has not been explored in prior studies related to return prediction; therefore, in our research, we trained models with and without the automated feature engineering technique to assess its impact on performance.

### 3.5 The proposed method

The baseline return prediction model used in this study is formulated as

$$Y_{t+1} = f(\mathbf{X}_t) = f(X_{1,t}, X_{2,t}, \dots, X_{12,t}), \tag{4}$$

where $f$ is a machine learning model, $Y_{t+1}$ denotes the return at quarter $t+1$, and $\mathbf{X}_t = \langle X_{1,t}, X_{2,t}, \dots, X_{12,t} \rangle$ represents the set of financial indicators from a company's financial statements at quarter $t$.

The extended return prediction model, incorporating DEA (Data Envelopment Analysis) efficiency scores, is formulated as

$$Y_{t+1} = f(\mathbf{X+}_t) = f(X_{1,t}, X_{2,t}, \dots, X_{12,t}, H_t), \tag{5}$$

where, $\mathbf{X+}_t$ is the extended feature set, and $H$ denotes the efficiency score of a specific company at quarter $t$.

Similarly, let $\mathbf{X+'}_t$ denote the feature-engineered versions of $\mathbf{X+}_t$. We then construct the following equation.

$$Y_{t+1} = f(\mathbf{X+'}_t) = f\big(g\left(X_{1,t}, X_{2,t}, \ldots, X_{12,t}, H_t\right)\big), \tag{6}$$

The machine learning models employed include GLM, DNNs, DT, RF, and GBT. The dataset is divided into training (2019?2023) and test (2024) sets. The training set is further split into 3 folds for model training, hyper-parameter tuning, and feature engineering (if applicable) using $k$-fold cross-validation. Finally, model performance is evaluated on the test set using criteria such as RMSE, MAE, and MAPE.

Through these models, the study seeks to answer the following research questions:

Research Question 1: Does incorporating efficiency scores as an input feature improve the performance of ML models? In other words, we compare the prediction errors of ML models on the test set when applying Eq (4) versus Eq (5).

Research Question 2: Does the combination of efficiency scores ($H$) and feature engineering further enhance ML model performance? This involves comparing the prediction errors of models using Eq (6) against the other formulations.

Research Question 3: What economic insights can be derived from the best-performing model regarding return prediction?

The workflow is summarized in Fig 3.

## 4 Result and discussion

### 4.1 $k$-Folds cross validation

The $k$-folds cross-validation process, specifically $k = 3$, is used to tune the hyper-parameters of the models and optimize the features in the case of the model using feature engineering techniques. Specifically, to tune the hyper-parameters, we perform optimization using a grid search in Altair AI Studio to optimize the correlation index between the predicted results and the actual results over the three folds. For each model, we select some essential hyper-parameters to optimize and keep the others at default. The investigated ranges and steps for the tuned parameters are as follows.

- GLM: lambda ($\lambda$) ranges from 0.0 to 1 with a step of 0.1.
- DANN: learning rate ($Lr$) ranges from 0.01 to 1 with a step of 0.1.
- GBT: max-depth ($m$) ranges from 1 to 10 with a step of 1;
- SVR: $C$ ranges from 0 to 1000 with a step of 100; $\gamma$ ranges from 0 to 1 with a step of 0.5;
- DT: $m$ ranges from 1 to 10 with a step of 1.

Meanwhile, the features are selected through a multi-objective optimization algorithm, with two goals: minimizing the RMSE over the three folds and minimizing the complexity of

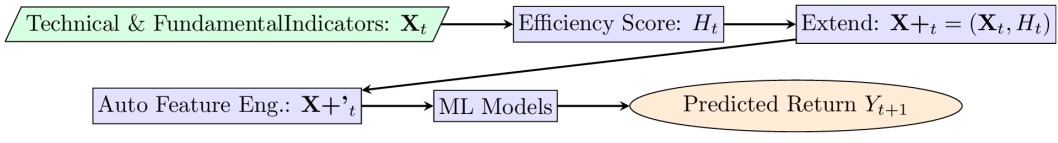

**Fig 3. The proposed method work-flow.**

the obtained features. The results for tuning parameters and feature engineering are recorded in Table 3.

## 4.2 Results on test set

The models with tuned hyper-parameters were evaluated on a test set, an extended dataset in 2024, which was not involved in the training process. The results obtained are shown in Table 4.

For question 1, Table 4 shows that using the efficiency score $H$ calculated from DEA can improve the performance of formula-based models such as DANN, and SVR, but not for rule-based models such as RF, and GBT. Specifically, the RMSE, MAE, and MAPE values of DANN are reduced from 0.926, 0.337, and 134.63 to 0.375, 0.196, and 114.71, respectively, when using the variable $H$. These results are very significant when there is a large difference between the error value of the original model and the error value of the improved model. Similarly, for SVR, the RMSE, MAE, and MAPE values decreased from 0.472, 0.398, and 109.04 to 0.236, 0.185, and 105.88, respectively, when using the $H$ variable. For tree-like

**Table 3**. **Results of hyper-parameter tuning and feature engineering.**

| Method | Hyper-parameters | Features |
|---|---|---|
| GLM | $\lambda$=0.7 | Original features |
| DEA-GLM | $\lambda$=0.7 | Original features, $H$ |
| F-DEA-GLM | $\lambda$=0.1 | log([$H$]) |
| DANN | $Lr$=0.1 | Original features |
| DEA-DANN | $Lr$=0.1 | Original features, $H$ |
| F-DEA-DANN | $Lr$=0.1 | log([$H$]) |
| SVR | $C$=0, $\gamma$=0.5 | Original features |
| DEA-SVR | $C$=1000, $\gamma$=0.5 | Original features, $H$ |
| F-DEA-SVR | $C$=100, $\gamma$=0.5 | log([$H$]) |
| RF | Number of trees=100, $m$=3 | Original features |
| DEA-RF | Number of trees=100, $m$=2 | Original features, $H$ |
| F-DEA-RF | Number of trees=100, $m$=5 | $H$, sgn($H$), max($H$, sgn($H$)) |
| GBT | Number of trees=50, $m$=9 | Original features |
| DEA-GBT | Number of trees=50, $m$=2 | Original features, $H$ |
| F-DEA-GBT | Number of trees=50, $m$=5 | sqrt(sqrt([$H$]))[Quarter], Pretax Margin |

**Table 4**. **The average performance of the test set.**

| Method | RMSE | MAE | MAPE |
|---|---|---|---|
| GLM | 0.155 | **0.129** | **117.30** |
| DEA-GLM | 0.155 | **0.129** | **117.30** |
| F-DEA-GLM | **0.154** | **0.129** | 118.81 |
| DANN | 0.926 | 0.337 | 134.63 |
| DEA-DANN | 0.375 | 0.196 | **114.71** |
| F-DEA-DANN | **0.182** | **0.153** | 126.06 |
| SVR | 0.472 | 0.398 | 109.04 |
| DEA-SVR | 0.236 | 0.185 | **105.88** |
| F-DEA-SVR | **0.155** | **0.130** | 117.62 |
| RF | **0.155** | 0.132 | 121.42 |
| DEA-RF | 0.16 | 0.134 | 118.34 |
| F-DEA-RF | 0.159 | **0.132** | **116.80** |
| GBT | **0.158** | 0.131 | 115.68 |
| DEA-GBT | 0.159 | 0.133 | 116.85 |
| F-DEA-GBT | **0.158** | **0.122** | **103.19** |

models, the error increased when using the *H* variable, but this increase was not significant; for example, the MAE of DEA-GBT was 0.133 compared to 0.131 of GBT. Thus, the results partly demonstrate the effectiveness of using the *H* variable, especially in the case of the original models built on mathematical formulas, which gave quite large errors compared to the tree-like models in the problem under consideration.

For question 2, we can see the effectiveness of feature engineering when the MAE of the feature engineering models all achieve the best results compared to themselves when using the original variables and using the additional variable H. In particular, the F-DEA-GBT model achieves the best overall results, in terms of MAE and MAPE, with values of 0.122 and 103.19, respectively. Fig 4 depicts the scatter plot of actual and predicted returns using F-DEA-GBT. It can be observed that the actual returns exhibit a positive correlation with the predicted returns. Higher predicted returns correspond to higher actual returns occurring subsequently. The specific value for the correlation coefficient in this case is 0.422. In our opinion, this value is acceptable for problems related to the financial domain, which often involve significant uncertainty. In terms of RMSE, the model achieving the best overall results is F-DEA-GLM with RMSE = 0.155. These results show, on the one hand the effectiveness of the feature engineering technique, and on the other hand, the contribution of the variable *H*, when the features used by F-DEA-GBT and F-DEA-GLM all have the presence of the variable H (see Table 3).

## 4.3 Discussion

The F-DEA-GBT model demonstrates the highest overall effectiveness on the test set (a Python code version of this model can be found at https://www.kaggle.com/code/thaonguyentrang/lgbm-feature-engineering-for-return-prediction). Therefore, it is selected for further in-depth analysis for Question 3. Through this model, we aim to uncover meaningful economic insights.

According to the model, two variables strongly influence the prediction outcome, including $sqrt(sqrt([H]))[Quarter]$ and *PretaxMargin* (Fig 5). Among these, the variable $sqrt(sqrt([H]))^*[Quarter]$ has the strongest impact. This indicates that the relationship between H and the predicted value of Return is quite complex, as it depends on the quarter.

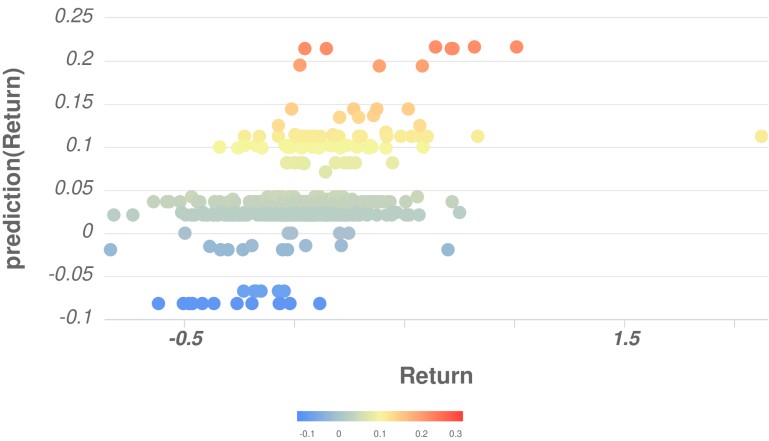

**Fig 4. The scatter plot of actual and predicted returns using F-DEA-GBT.**

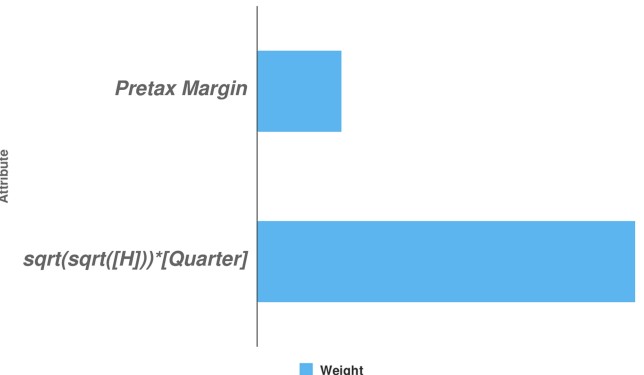

**Fig 5. The degree of the impact of variables on predictive outcomes.**

Therefore, it is necessary to consider the interaction between *H* and *Quarter* when predicting Return.

First, the paper examines the relationship between the Quarter and the predicted Return (Fig 6). It can be observed that there are differences in Return across quarters. Return tends to remain relatively stable in Q2 and Q4, often reaching higher values in Q3, while exhibiting significant volatility in Q1. In our view, this result is quite reasonable, as Q1 coincides with both the Gregorian and Lunar New Year holidays in the Vietnamese market. During this period, investors typically become more cautious, leading to the selling of stocks to recover capital, which results in price declines and heightened volatility. Trading volumes also tend to decrease as many investors take time off, making stock prices more susceptible to the influence of large transactions.

According to the F-DEA-GBT model, in addition to *H* and *Quarter*, *Returnprediction* is also influenced by *PretaxMargin*. Fig 7 illustrates the nonlinear relationship between these two variables. The *Returnprediction* may turn negative when the *PretaxMargin* approaches zero. Conversely, the average predicted return tends to increase with the magnitude of *PretaxMargin*, regardless of whether this value is negative or positive.

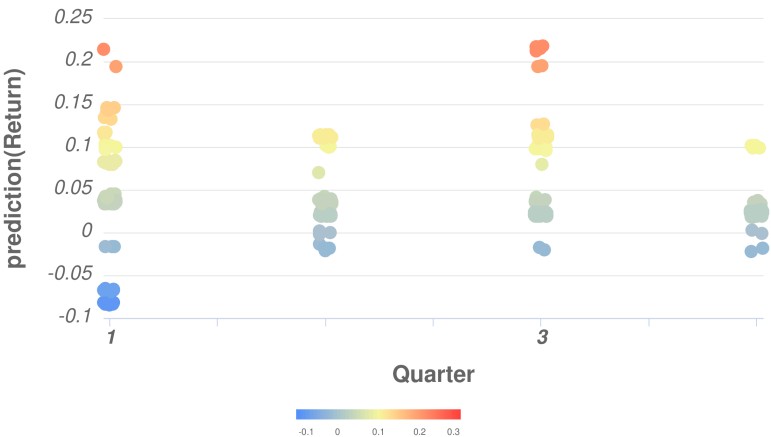

**Fig 6. The relationship between Return and sqrt(sqrt([H]))*[Quarter].**

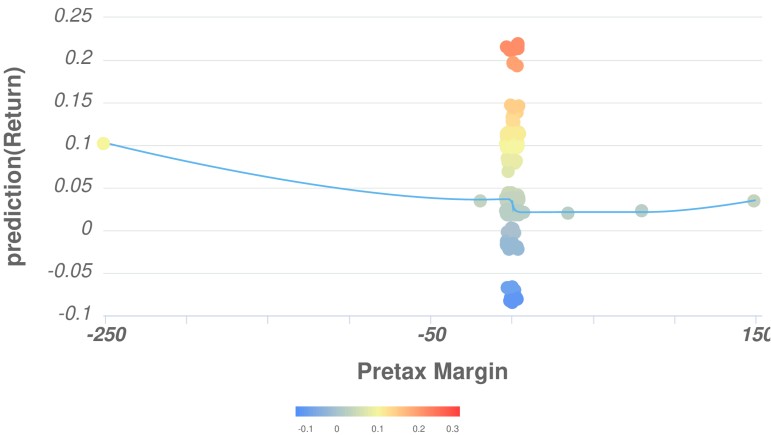

**Fig 7. The relationship between Return Prediction and Pretax Margin.**

Fig 7 describes a nonlinear relationship that appears to exist. The trend line on the chart exhibits a slight curve, indicating a nonlinear relationship between *Returnprediction* and *PretaxMargin*. At low levels of *PretaxMargin* (near -250), *Returnprediction* tends to be positive but increases only slightly. In the middle range of the axis (approximately -50 to 50), the *Returnprediction* remains almost unchanged or fluctuates slightly around zero. At higher levels of *PretaxMargin* (above 50), *Returnprediction* shows a slight upward trend, though not significantly. Based on this nonlinear relationship, we offer the following explanation.

When the *PretaxMargin* is higher, the stock prices of real estate companies tend to rise accordingly. This suggests that investors have greater confidence in companies that focus on their core business operations rather than diversifying investments, as such companies are likely to generate stable and sustainable revenue streams. For example, Nam Long Real Estate Corporation (NLG) concentrates on developing housing for middle-income earners and avoids investing in resort projects, which allows it to maintain consistently high *PretaxMargins*. As a result, NLG's stock price has shown steady and stable growth.

When the *PretaxMargin* is low, there begins to be a slight upward trend in the stock prices of real estate companies. This reflects a modest level of investor confidence that large real estate firms, such as Novaland (NVL) and Hung Thinh (HTN), will recover effectively following a period of corporate restructuring.

## 5 Conclusion

This paper has demonstrated that integrating DEA scores with automatic feature engineering significantly enhances prediction accuracy. Among the models tested, the F-DEA-GBT model, which combines DEA-based features with the Gradient Boosted Tree algorithm, achieved the best overall performance. From an academic perspective, the study offers several notable contributions. It is among the first to incorporate DEA efficiency scores into stock return prediction and to apply automatic feature engineering to panel data. These findings extend the current body of knowledge on emerging markets like Vietnam by demonstrating the value of efficiency-based indicators and advanced machine learning techniques in improving forecast accuracy. However, the current paper has some limitations: the sample data is small, and the Vietnamese Stock Market is idiosyncratic. Focusing solely on 26 real estate companies listed on the Ho Chi Minh Stock Exchange reduces the generalizability of results to other sectors or broader markets. Therefore, findings derived from this sample may not

reflect the behavior or return patterns of firms in manufacturing, technology, or services. The narrow dataset also limits the statistical power of machine learning models and increases the risk of model overfitting, particularly when using complex algorithms. Additionally, the Vietnamese market is young and developing, with low liquidity compared to mature markets. It is strongly influenced by seasonal factors, such as the Lunar New Year, which affects investor sentiment and trading volumes, especially in the first quarter. Therefore, this unique characteristic reduces the transferability of machine learning models trained in other contexts. Future research can build on these findings by exploring additional variables and incorporating broader datasets to further refine predictive accuracy in diverse market contexts.

## Author contributions

**Conceptualization:** Hoang Thanh Nhon, Thao Nguyen-Trang.

**Data curation:** Hoang Thanh Nhon.

**Formal analysis:** Thao Nguyen-Trang.

**Investigation:** Hoang Thanh Nhon.

**Methodology:** Hoang Thanh Nhon, Thao Nguyen-Trang.

**Project administration:** Hoang Thanh Nhon.

**Supervision:** Thao Nguyen-Trang.

**Validation:** Thao Nguyen-Trang.

**Visualization:** Thao Nguyen-Trang.

**Writing – original draft:** Hoang Thanh Nhon, Nga Do-Thi, Thao Nguyen-Trang.

**Writing – review & editing:** Nga Do-Thi.

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
