## [Decision Letter · Decision Letter 0]

2 Jul 2025

PONE-D-25-32672Predicting stock returns using machine learning combined with data envelopment analysis and automatic feature engineering: A case study on the Vietnamese stock marketPLOS ONE

Dear Dr. Nguyen-Trang,

Thank you for submitting your manuscript to PLOS ONE. After careful consideration, we feel that it has merit but does not fully meet PLOS ONE’s publication criteria as it currently stands. Therefore, we invite you to submit a revised version of the manuscript that addresses the points raised during the review process.

We look forward to receiving your revised manuscript.

Kind regards,

Miguel Alves Pereira

Academic Editor

PLOS ONE

Journal Requirements:

2. In the online submission form, you indicated that “The data used in this study are available upon request. If you require access to the data, please contact the corresponding author.”

**Additional Editor Comments:**

The manuscript presents a methodologically sound and potentially valuable contribution to the intersection of efficiency analysis and machine learning for stock return prediction in emerging markets. However, the manuscript requires further refinement to meet publication standards. Specifically, the abstract and conclusion fail to adequately convey the study’s impact, and the presentation of results (particularly in terms of model comparisons and hyperparameter settings) lacks the transparency necessary for reproducibility. The limitations of the study, especially the small and sector-specific sample, must be explicitly acknowledged. Figures should be revised to better reflect the proposed approach, and concrete examples of feature engineering outputs would improve clarity.

Reviewers' comments:

Reviewer's Responses to Questions

**Comments to the Author**

1. Is the manuscript technically sound, and do the data support the conclusions?

Reviewer #1: Yes

Reviewer #2: Partly

2. Has the statistical analysis been performed appropriately and rigorously? 

Reviewer #1: Yes

Reviewer #2: No

3. Have the authors made all data underlying the findings in their manuscript fully available?

Reviewer #1: Yes

Reviewer #2: Yes

4. Is the manuscript presented in an intelligible fashion and written in standard English?

Reviewer #1: Yes

Reviewer #2: No

5. Review Comments to the Author

Reviewer #1: The manuscript titled "Predicting stock returns using machine learning combined with data envelopment analysis and automatic feature engineering: A case study on the Vietnamese stock market" presents an original and well-structured approach to stock return prediction in an emerging market. Its main contribution lies in the use of efficiency scores derived from Data Envelopment Analysis (DEA) as explanatory variables within machine learning models — a technique still underexplored in financial prediction literature.

The manuscript is technically sound, with a solid methodology and appropriate validation via 3-fold cross-validation. The authors applied various machine learning models (GLM, DANN, SVR, Random Forest, and GBT), with hyperparameter tuning conducted via grid search. The inclusion of automatic feature engineering further enhances the study, revealing significant improvements in error metrics (RMSE, MAE, MAPE).

Strengths

- Originality: The integration of DEA efficiency scores is novel, especially in the context of the Vietnamese stock market.

- Robust methodology: Combining technical, fundamental, and efficiency variables with automated feature engineering strengthens model performance.

- Practical relevance: Focusing on an emerging market with low liquidity and political sensitivity adds value for local analysts and investors.

- Clarity: The manuscript is well organized, and the modeling steps are clearly explained.

Points for Improvement

- Limitations discussion: The authors are encouraged to include a specific section discussing the study's limitations, such as the small sample (26 real estate firms) and the idiosyncrasies of the Vietnamese market.

- Feature engineering details: The general explanation is appropriate, but a concrete example of a transformed variable or a table of generated features would enhance clarity.

- Additional visualizations: Graphs comparing predicted vs. actual returns would help readers assess the model's real-world performance.

- Data availability: While the authors state that data are fully available, providing a direct link to a public repository would improve transparency.

Reviewer #2: The paper addresses a good research gap, but due to presentation it lacks the required amount of impact.

1. Abstract does not provided impactive idea regarding the final result. A little summary regarding results will be helpful.

2. Introduction may be shortened and made more technical, a good part of it may be transferred to literature review.

3. A summary of contribution of the paper in listed manner may be added. Paragraph-wise explanation is good, but a summary list will help the reader to understand the paper's content most effectively.

4. Though it is told where to find data, either providing a repository containing it, or giving some lead on how to re-generate it will help future researchers to explore your idea more. Try adding links and guidelines to data procurement or generation more elaboratively.

5. Figures (Diagrams) should be improved to make everything look visually more sound. Figure 3 is just a general diagram of how machine learning works, there is not much to relate to specific proposed model.

6. Comparison between results from different models should be presented more informatively. Hyperparameters could be shown more systematically. Overall, recreating the results of the paper will help future researches generated from it. So more information regarding models and parameters should be added.

7. Conclusion (just like abstract) does hold the full impact of the work. It should be re-written to be more concise and informative regrading the work done.

6. PLOS authors have the option to publish the peer review history of their article (what does this mean?). If published, this will include your full peer review and any attached files.

Reviewer #1: No

Reviewer #2: **Yes: **Tashreef Muhammad

---

## [Author Response · Author response to Decision Letter 1]

18 Aug 2025

Authors’ Responses to Reviewers

Firstly, we deeply appreciate the advice and encouragement from the Editors and the Reviewers. We thank the Reviewers for their careful evaluation of the manuscript. We have revised the paper with respect to the Reviewer’s comments. We hope that our revised paper will meet the Reviewers’ requirements. The point-by-point responses to the Editors’ and Reviewers’ comments can be found below. It should be noted that the revision is illustrated by the blue color in the revised manuscript.

Additional Editor Comments:

The manuscript presents a methodologically sound and potentially valuable contribution to the intersection of efficiency analysis and machine learning for stock return prediction in emerging markets. However, the manuscript requires further refinement to meet publication standards. Specifically, the abstract and conclusion fail to adequately convey the study’s impact, and the presentation of results (particularly in terms of model comparisons and hyperparameter settings) lacks the transparency necessary for reproducibility. The limitations of the study, especially the small and sector-specific sample, must be explicitly acknowledged. Figures should be revised to better reflect the proposed approach, and concrete examples of feature engineering outputs would improve clarity.

Response: Yes, thank you for the Editors' comments. In the revised manuscript, we have discussed the study's limitations in the conclusion. Additionally, we have clarified the methodology by adding new paragraphs related to feature engineering operators and the search space of parameters. The data and code have also been provided via links. Furthermore, the abstract and conclusion have been revised to better highlight the impacts and results of the study. Finally, the English has been double-checked and corrected. For detailed information, please refer to our responses to the specific comments from the Reviewers below.

Reviewer #1 Comments

- Limitations discussion: The authors are encouraged to include a specific section discussing the study's limitations, such as the small sample (26 real estate firms) and the idiosyncrasies of the Vietnamese market.

Response: Yes, thank you for the Reviewer’s comment. We fully agree that the current paper has some limitations: the sample data is small, and the Vietnamese stock market is idiosyncratic. We have added the following discussion to the conclusion section. Firstly, the proposed method is used in predicting the return of real estate companies listed on the Ho Chi Minh Stock Exchange. This could reduce the generalizability of results to other sectors or broader markets. Therefore, findings derived from this sample may not reflect the behavior or return patterns of firms in manufacturing, technology, or services. The narrow dataset also limits the statistical power of machine learning models and increases the risk of model overfitting, particularly when using complex algorithms. Additionally, the Vietnamese market is young and developing, with low liquidity compared to mature markets. It is strongly influenced by seasonal factors. For example, the Lunar New Year could affect investors’ sentiment and trading volumes, especially in the first quarter. Therefore, this unique characteristic might reduce the transferability of machine learning models trained in different contexts. The corresponding changes can be found in the revised manuscript, lines 387-399.

- Feature engineering details: The general explanation is appropriate, but a concrete example of a transformed variable or a table of generated features would enhance clarity.

Response: Yes, in response to the Reviewer’s comments, we have included further details of the functions used. The corresponding changes can be seen in the revised manuscript, line 241, and in the new Table 2.

- Additional visualizations: Graphs comparing predicted vs. actual returns would help readers assess the model's real-world performance.

Response: Yes. We appreciate the Reviewer’s comment. We have added a new figure to depict the scatter plot of actual and predicted returns using F-DEA-GBT. As shown in the new Fig. 4 in the revised manuscript, the actual returns exhibit a positive correlation with the predicted returns. Higher predicted returns correspond to higher actual returns occurring subsequently. The specific value for the correlation coefficient in this case is 0.422. In our opinion, this value is acceptable for issues related to the financial domain, which often involve significant uncertainty. The corresponding changes can be found in the revised manuscript, specifically in the new Fig. 4 and lines 320-325.

- Data availability: While the authors state that data are fully available, providing a direct link to a public repository would improve transparency.

Response: Yes, thank you for the Reviewer’s comment. For reference, we have published the detailed dataset at the following link: “https://www.kaggle.com/datasets/thaonguyentrang/vietnam-stock-returns-with-efficiency-scores-dea”. The corresponding changes can be found in the revised manuscript, from line 143 to line 146.

Reviewer #2 Comments

1. Abstract does not provided impactive idea regarding the final result. A little summary regarding results will be helpful.

Response: Yes. We appreciate the Reviewer’s comments. We have added a new paragraph to the abstract: “The results indicate that incorporating business efficiency scores significantly enhances the models' accuracy. For example, the deep neural network model shows a decrease in RMSE from 0.926 to 0.375, MAE from 0.337 to 0.196, and MAPE from 134.63 to 114.71. Furthermore, the gradient boosted tree model, when combined with business efficiency scores and automatic feature engineering, achieves the best results, yielding an MAE of 0.122 and a MAPE of 103.19”.

2. Introduction may be shortened and made more technical, a good part of it may be transferred to literature review.

Response: We appreciate the Reviewer’s comments. We have restructured the introduction and literature review sections for improved coherence. Additionally, the language has been refined for greater technical precision. Detailed revisions can be found in the revised manuscript, specifically on lines 1-26 and 34-130.

3. A summary of contribution of the paper in listed manner may be added. Paragraph-wise explanation is good, but a summary list will help the reader to understand the paper's content most effectively.

Response: Yes. Thanks for the Reviewer’s comments. We have included a paragraph summarizing the paper’s contributions. Particularly,

- This paper is one of the first studies that considers the efficiency score calculated by DEA as a predictor for stock return prediction.

- This paper pioneers the application of automatic feature engineering to explore latent and essential predictors related to the efficiency score and other variables, thereby enhancing the model's performance.

- This paper serves as a useful reference for selecting an appropriate machine learning model to predict stock returns in the Vietnamese market based on the efficiency score.

The corresponding changes can be seen in the revised manuscript, lines 19-26.

4.Though it is told where to find data, either providing a repository containing it, or giving some lead on how to re-generate it will help future researchers to explore your idea more. Try adding links and guidelines to data procurement or generation more elaboratively.

Response: Yes, thank you for the Reviewer’s comment. For reference, we have published the detailed dataset at the following link: “https://www.kaggle.com/datasets/thaonguyentrang/vietnam-stock-returns-with-efficiency-scores-dea”. The corresponding changes can be found in the revised manuscript, from line 143 to line 146.

5. Figures (Diagrams) should be improved to make everything look visually more sound. Figure 3 is just a general diagram of how machine learning works, there is not much to relate to specific proposed model.

Response: Yes, thank you for the Reviewer’s comment. We have revised the flowchart in Fig. 3 to better reflect the main idea of the proposed algorithm. Please refer to the updated Fig. 3 in the revised manuscript.

6. Comparison between results from different models should be presented more informatively. Hyperparameters could be shown more systematically. Overall, recreating the results of the paper will help future researches generated from it. So more information regarding models and parameters should be added.

Response: Yes. In the revised manuscript, we have added paragraphs related to automatic feature engineering, hyper-parameter tuning, and a link to the Python code. The details are as follows.

We have introduced a new Table 2 to clarify the operators used in the automatic feature engineering process.

Additionally, we have specified the search space for the hyper-parameters of the models used. Specifically, for tuning the hyper-parameters of the used models, we perform the grid search optimization in Altair AI Studio to maximize the correlation index between the predicted results and the actual outcomes across the three folds. For each model, we select key hyper-parameters to optimize while keeping the others at their default values. The investigated ranges and steps for the tuned parameters are as follows:

oGLM: lambda ranges from 0.0 to 1 with a step of 0.1.

oDANN: learning rate ranges from 0.01 to 1 with a step of 0.1.

oGBT: max-depth ranges from 1 to 10 with a step of 1;

oSVR: C ranges from 0 to 1000 with a step of 100; gamma ranges from 0 to 1 with a step of 0.5;

oDT: m ranges from 1 to 10 with a step of 1.

We have provided a link containing the Python code for the best model to ensure repeatability: https://www.kaggle.com/code/thaonguyentrang/lgbm-feature-engineering-for-return-prediction

The corresponding changes can be found in the revised manuscript, lines 282-292, 331-334, the new Table 2, and Table 3.

7. Conclusion (just like abstract) does hold the full impact of the work. It should be re-written to be more concise and informative regrading the work done.

Response: Yes. We appreciate the Reviewer’s comments. We have rewritten the conclusion to better reflect the results and impact of the study: “This paper has demonstrated that integrating DEA scores with automatic feature engineering significantly enhances prediction accuracy. Among the models tested, the F-DEA-GBT model, which combines DEA-based features with the Gradient Boosted Tree algorithm, achieved the best overall performance. From an academic perspective, the study offers several notable contributions. It is among the first to incorporate DEA efficiency scores into stock return prediction and to apply automatic feature engineering to panel data. These findings extend the current body of knowledge on emerging markets like Vietnam by demonstrating the value of efficiency-based indicators and advanced machine learning techniques in improving forecast accuracy. However, the current paper has some limitations: the sample data is small, and the Vietnamese Stock Market is idiosyncratic. Focusing solely on 26 real estate companies listed on the Ho Chi Minh Stock Exchange reduces the generalizability of results to other sectors or broader markets. Therefore, findings derived from this sample may not reflect the behavior or return patterns of firms in manufacturing, technology, or services. The narrow dataset also limits the statistical power of machine learning models and increases the risk of model overfitting, particularly when using complex algorithms. Additionally, the Vietnamese market is young and developing, with low liquidity compared to mature markets. It is strongly influenced by seasonal factors, such as the Lunar New Year, which affects investor sentiment and trading volumes, especially in the first quarter. Therefore, this unique characteristic reduces the transferability of machine learning models trained in other contexts. Future research can build on these findings by exploring additional variables and incorporating broader datasets to further refine predictive accuracy in diverse market contexts”. The corresponding changes can be seen in the revised manuscript, lines 379-399.

---

## [Decision Letter · Decision Letter 1]

27 Aug 2025

Predicting stock returns using machine learning combined with data envelopment analysis and automatic feature engineering: A case study on the Vietnamese stock market

PONE-D-25-32672R1

Dear Dr. Nguyen-Trang,

We’re pleased to inform you that your manuscript has been judged scientifically suitable for publication and will be formally accepted for publication once it meets all outstanding technical requirements.

Kind regards,

Miguel Alves Pereira

Academic Editor

PLOS ONE

Additional Editor Comments (optional):

Reviewers' comments:

Reviewer's Responses to Questions

**Comments to the Author**

1. If the authors have adequately addressed your comments raised in a previous round of review and you feel that this manuscript is now acceptable for publication, you may indicate that here to bypass the “Comments to the Author” section, enter your conflict of interest statement in the “Confidential to Editor” section, and submit your "Accept" recommendation.

Reviewer #2: All comments have been addressed

2. Is the manuscript technically sound, and do the data support the conclusions?

Reviewer #2: Yes

3. Has the statistical analysis been performed appropriately and rigorously? 

Reviewer #2: Yes

4. Have the authors made all data underlying the findings in their manuscript fully available?

Reviewer #2: Yes

5. Is the manuscript presented in an intelligible fashion and written in standard English?

Reviewer #2: Yes

6. Review Comments to the Author

Reviewer #2: I think the authors did a good job addressing the issues as enlisted. A little suggestion from me will be to include the links of kaggle as footnotes rather than writing them as text in the main body of the paper.

7. PLOS authors have the option to publish the peer review history of their article (what does this mean?). If published, this will include your full peer review and any attached files.

Reviewer #2: **Yes: **Tashreef Muhammad

---

## [Editor Report · Acceptance letter]

PONE-D-25-32672R1

PLOS ONE

Dear Dr. Nguyen-Trang,

I'm pleased to inform you that your manuscript has been deemed suitable for publication in PLOS ONE. Congratulations! Your manuscript is now being handed over to our production team.

Kind regards,

on behalf of

Prof. Miguel Alves Pereira

Academic Editor

PLOS ONE